# Accurate Step Count with Generalized and Personalized Deep Learning on Accelerometer Data [note 1]

**DOI:** 10.3390/s22113989

**Published:** 2022-05-24

**Authors:** Long Luu, Arvind Pillai, Halsey Lea, Ruben Buendia, Faisal M. Khan, Glynn Dennis

**Affiliations:** 1Digital Health, Oncology R&D, AstraZeneca, Gaithersburg, MD 20878, USA; halsey.lea@astrazeneca.com; 2Department of Computer Science, Dartmouth College, Hanover, NH 03755, USA; arvind.pillai184@gmail.com; 3Biometrics, Late-Stage Development, Cardiovascular, Renal and Metabolism (CVRM), BioPharmaceuticals R&D, AstraZeneca, 43183 Gothenburg, Sweden; ruben.buendia@astrazeneca.com; 4AI & Analytics, Data Science & Artificial Intelligence R&D, AstraZeneca, Gaithersburg, MD 20878, USA; faisalmkhan@gmail.com (F.M.K.); glynnsc@gmail.com (G.D.)

**Keywords:** deep learning, step count, accelerometer, wearable, healthcare, medicine, bioinfomatics

## Abstract

Physical activity (PA) is globally recognized as a pillar of general health. Step count, as one measure of PA, is a well known predictor of long-term morbidity and mortality. Despite its popularity in consumer devices, a lack of methodological standards and clinical validation remains a major impediment to step count being accepted as a valid clinical endpoint. Previous works have mainly focused on device-specific step-count algorithms and often employ sensor modalities that may not be widely available. This may limit step-count suitability in clinical scenarios. In this paper, we trained neural network models on publicly available data and tested on an independent cohort using two approaches: generalization and personalization. Specifically, we trained neural networks on accelerometer signals from one device and either directly applied them or adapted them individually to accelerometer data obtained from a separate subject cohort wearing multiple distinct devices. The best models exhibited highly accurate step-count estimates for both the generalization (96–99%) and personalization (98–99%) approaches. The results demonstrate that it is possible to develop device-agnostic, accelerometer-only algorithms that provide highly accurate step counts, positioning step count as a reliable mobility endpoint and a strong candidate for clinical validation.

## 1. Introduction

Physical activity (PA) is well established as a fundamental measure of general health and wellness. In fact, in 2008 the US government released its first-ever evidence-based guidance on PA, which acknowledged very strong evidence that physically active people have higher levels of health-related fitness, a lower risk profile for developing cardiovascular diseases and lower rates of various chronic diseases than people who are not active [1]. A more recent systematic review has shown that PA improves several health outcomes, including all-cause mortality and cardiovascular diseases [2]. More recent works also show the benefit of combining PA with other clinically relevant measures, such as temperature, to gain insight into a patient’s health status [3]. As a measure of PA, step counts derived from digital devices have gained increasing popularity as an objective measure to replace self-reported data [4,5,6]. An emerging body of literature has shown robust association between step count and several health outcomes, including all-cause mortality, cardiovascular disease, diabetes and obesity [7,8,9,10,11]. A similar pattern has also been found in specific clinical populations, such as the elderly [12,13,14] and people with impaired glucose tolerance [15]. Moreover, results from interventional studies show beneficial effects of increasing step count, such as weight loss [16] and several long-term health outcomes [17].

Despite conclusive evidence of the medical relevance of step counts, such objective measures of daily physical activity have yet to be established as accepted clinical endpoints or biomarkers. The main reason is that, although many types of wearables have been used in both clinical and non-clinical settings to measure step count, in most cases the algorithms used with those devices have not been rigorously and extensively validated [18]. Although previous studies have found high step-count accuracy for healthy subjects in both lab-controlled [19] and free-living conditions [20], the accuracy is often compromised for elderly and clinical populations [21,22,23,24]. The reason could be because those subjects have idiosyncratic gaits. Along the same line, previous studies also showed that step-count devices may not work well in non-regular walking conditions, such as slow walking or pushing a stroller [25]. As a result, there is a pressing need to develop algorithms that can work well across different subject populations and devices.

Broadly, step-counting algorithms can be grouped based on detection mechanisms— time domain approaches, frequency domain approaches and machine learning/deep learning approaches. Thresholding is a common time domain approach which detects a step when the sensor data meet pre-specified criteria [26]. Peak detection is another popular approach which detects a step when the peaks of an acceleration signal exceed a certain threshold [27]. Other works employ different techniques, such as auto-correlation [28] and template matching with dynamic time warping [29] on the time series to detect a step. In general, frequency domain approaches utilize features in the frequency domain to detect a step [30]. In an extensive benchmark study, two frequency-based approaches were used: short-term Fourier transform (STFT) and continuous/discrete wavelet transform (CWT/DWT) [31]. The STFT approach applied STFT to successive windows of acceleration to compute a fractional number of strides in each window. The CWT/DWT method transformed the data with CWT/DWT, extracted the walking periods from the energy spectrum, then transformed it back to the time domain and used positive zero-crossing to count steps.

Although the above algorithms could produce good results for the step-counting task, they often require careful tuning of several parameters for each situation. To alleviate that issue, machine learning models have been used to automatically learn those parameters with cross validation. A popular choice, the Hidden Markov Model (HMM), aimed to predict different phases of walking with hidden states and to count steps based on those phases [31,32]. Along the same line, k-means clustering has also been used to compute two clusters (hill/valley) from a rolling time window of the feature vector, and the step count was computed from that [31]. A recent work used a k-nearest neighbor algorithm to learn the optimal parameters for extracting the most relevant and generalizable features in both time and frequency domains [33].

Despite several successes with traditional step-count approaches, the necessity for expert-based hand-tuning of useful features remains a barrier to standardization or generalization. Contemporary deep learning approaches have shown superior predictive power and classification accuracy across numerous domains in visual, acoustic, linguistic, gaming and scientific domains [34]. This superior performance is due, in part, to deep learning models’ inherent ability to learn and engineer predictive features from raw data. There have been recent attempts to utilize deep learning models for step detection [35,36,37]. Those studies utilized state-of-the-art deep learning architectures, such as LSTM recurrent network and convolutional neural network, and show promising results (accuracy above 90%). However, because neural network models have very high expressive power, they can easily overfit a relatively small dataset, such as those used in step-count research.

To address the overfitting problem of neural network models, recent work applied domain adaptation to perform transfer learning from a public dataset to an independently collected set of data [38]. Specifically, recurrent neural networks with LSTM cells were trained on a public cohort with 30 subjects and then adapted to an independent cohort of 5 subjects. Very high step-count accuracy was achieved (∼99%) using both accelerometer and gyroscope signals along with domain adaptation on 30 s labeled data for each subject. Although highly accurate, the work in [38] presented three limitations that can hinder wide application of the transfer learning methodology in clinical scenarios. First, the model required both the accelerometer and gyroscope to achieve good results, and the model’s performance was significantly degraded with only accelerometer data (∼60%). Because many wearable devices only use the accelerometer in order to save battery life (especially in medical settings), application of the method could be quite limited in terms of devices. Second, to perform domain adaptation, the model required a small amount of labeled data for each individual, which could be quite burdensome for patients in clinical settings. Finally, the domain adaptation was limited to a single device, so it remains unclear how the algorithms and the approach generalize broadly for other devices.

In this work, we set out to address such limitations by: (1) collecting new data from subjects outfitted with four devices, (2) modifying the training process, and (3) extensively testing a wide array of state-of-the-art neural network models for time-series data. In particular, we experimented with a convolutional network, temporal convolutional network (WaveNet) and recurrent neural network. Here we tested two approaches: generalized and personalized transfer learning. The generalized approach used models pre-trained on a public dataset to directly test on an independent subject cohort wearing different devices without further parameter tuning. That approach achieved very high step-count accuracy across the majority of subjects and devices (96–99% with the best model). However, we also found that for some subjects and devices the performance was significantly degraded. Therefore, we implemented a second approach using domain adaptation to fine-tune the parameters of the pre-trained model with a small amount of labeled data for each individual subject. This approach resulted in significant improvement for the outlier subjects that the generalized method could not handle well. Therefore, a combination of the two approaches in practical applications can alleviate the expensive needs of large amount of labeled data while still preserving step-count accuracy.

## 2. Materials and Methods

### 2.1. Dataset

#### 2.1.1. Public Dataset

The public dataset was collected at Clemson University and consists of 30 subjects performing three different activities (regular walking, semi-regular walking and unstructured activity) while wearing Shimmer3 (Shimmer, Dublin, Ireland) devices at the hip, ankle and wrist [39]. For this study, we only used the regular walking activity, in which subjects walked at a comfortable pace around a building for approximately 10 min. The whole walking duration was recorded with an iPhone (Apple, Cupertino, CA, USA) for subsequent manual annotation. The annotation indicates whether each time point corresponded to a left or right step. For sensor data, we used the tri-axial accelerometer signal from the wrist location that was sampled at 15 Hz and had a dynamic range of ±2 g.

#### 2.1.2. Independent Dataset

The independent dataset was collected at AstraZeneca, Gaithersburg, MD, USA with a total of 11 subjects (5 subjects in [38] and 6 new subjects). They were instructed to walk at a regular pace around a looped path with minimum length of 30 m for 6 min. The activity mimicked the typical Six Minute Walk Test that is often used in clinical settings to assess the functional capacity of a patient. While walking, subjects wore an ActiGraph GT9X 130 Link (Actigraph, Pensacola, FL, USA) and an Apple Watch 6 (Apple, Cupertino, CA, USA) on their left wrist, a Shimmer Verisense (Shimmer, Dublin, Ireland) band on their right wrist and carried an iPhone 11 (Apple, Cupertino, CA, USA) in their left pocket.

ActiGraph wearable devices have been widely used in many clinical studies to measure physical activity. The version we tested, GT9X 130 Link, has a sampling rate of 50 Hz with three sensors (accelerometer, gyroscope and magnetometer). The dynamic ranges of the sensors are as follows: ±8 g for accelerometer, 132 ± 2000 deg/s for gyroscope and ±4800 micro-Tesla for magnetometer. The recorded data were stored locally and then uploaded to a cloud database via USB interface.

The Shimmer Verisense wearable device is another popular choice for clinical applications. The largest advantage of this device is that the battery life can be up to 6 months without any charge (compared to 2 days for the ActiGraph GT9X 130 Link and most other devices). The reason it has such a long battery life is that only the accelerometer is used. The data were sampled at 50 Hz with a dynamic range of ±8 g. Data were stored locally and automatically transferred to a base station, which periodically uploaded the data to a secure AWS S3 bucket.

To increase the diversity of tested devices, along with the above medical-grade wearables, we also tested two popular consumer devices: the Apple Watch 6 and iPhone 11. We used the iOS app “SensorLog” to capture raw sensor data (accelerometer, gyroscope and magnetometer) from both devices. The sampling rate of all data was set to 50 Hz. The Apple Watch also contained heart rate data obtained from the PPG sensor. Note that although most devices could capture a wide variety of sensor data, we only used accelerometer data for our algorithm because that is the only sensor available across all devices.

Similar to the public dataset, a video recording of the whole activity was used to annotate whether each time point corresponded to a left or right step. The recording was made with a Samsung (Samsung, Seoul, Korea) phone and an Android app “Timestamp Camera” that has the timestamp directly imposed on the bottom right corner of the video (see Figure 1). The experiment was conducted as follows. At the beginning, a subject stood still for 30 s so that the time series of all sensors could be aligned with each other in case the absolute timestamps did not work well. Then, they were asked to walk at a regular pace around a corridor that was at least 30 m long. After 6 min, the subject was asked to stop.

### 2.2. Data Preprocessing

The accelerometer signals in the independent dataset were resampled to match the 15 Hz sampling rate of the public dataset. Samples of tri-axial accelerometer signals from all devices are shown in Figure 2. The plot demonstrates a typical problem of applying an algorithm across different devices. The three axes of the accelerometers in the public dataset have different mean values and scales compared to those in the independent dataset. This is because accelerometer sensors from different devices often have different orientations relative to the earth’s gravity. Furthermore, the signal pattern from the iPhone is significantly different from the others mainly because it was placed in a different location (pant pocket instead of wrist). Therefore, we tried to use either the raw accelerometer data or the Euclidian Norm Minus One (ENMO) as the input into our model. The ENMO was computed as follows: $a_{ENMO}=\sqrt{a_{x}^{2}+a_{y}^{2}+a_{z}^{2}}-1$ where $a_{x},a_{y},a_{z}$ are raw acceleration signals corresponding to the three axes of the accelerometer sensors. As a result, the ENMO is a 1-dimensional measure as opposed to the 3-dimensional raw acceleration, and it is less sensitive to changes of the sensors’ axes. This is illustrated in Figure 3, in which the ENMO corresponding to raw sample data in Figure 2 was shown. Although it cannot resolve all issues with the raw acceleration, the ENMO measurements from different devices are more homogeneous than the raw acceleration data.

### 2.3. Neural Network Models

#### 2.3.1. Recurrent Neural Network with LSTM Cells

Recent successes in sequence data modeling have been attributed to the development of recurrent neural network models that naturally capture the temporal dependency in a time series with the hidden states of the network [40]. Different from the more popular feed-forward architectures such as convolution, recurrent networks have feedback connections so that each neural unit receives the past input from other units and itself. Therefore, they naturally take into account the temporal dependency between the current time point and past time points. Moreover, they can generally handle variable-length input well, which is an important feature for time-series data. However, the recurrent architecture often encounters the problem of vanishing/exploding gradients in error back-propagation during training. That significantly limits its practical applicability. Therefore, recurrent architecture only gained popularity with the development of modifications to the neural unit, such as long short-term memory (LSTM) [41] or gated recurrent unit (GRU) [42]. These variants help stabilize the training process and adaptively learn long-term dependency in the input sequence. They essentially utilize gated mechanisms to learn how much information from the past should be taken into account. Here we employ a typical recurrent network that consists of two layers of LSTM cells followed by two fully connected layers. The two LSTM layers have 256 and 128 cells, respectively, and the two fully connected layers have 128 and 2 cells, respectively. The last fully connected layer has a softmax activation function to predict the binary label in the dataset, i.e., left or right step. The LSTM layers employ tanh activation for the output and sigmoid activation for recurrent connection. The next-to-last fully connected layer employs ReLU activation. Besides the differences in number of layers and cells, one significant difference compared to previous work [38] is that the output of the last LSTM layer is the whole output sequence of the time series instead of just the output of the last time step. We believe that employing the output of the whole sequence will help make the back-propagation of the gradient to early time steps better than just the output of the last time step.

#### 2.3.2. Convolutional Neural Networks

Although recurrent networks might be a natural choice for sequence modeling, convolutional architecture as well has been proposed as a prominent alternative [43,44]. The biggest issue with recurrent networks is that, for sequence data with very high sampling rates it needs to learn over a very long sequence for potential dependency between time steps. For example, an audio signal with a typical 40 kHz sampling rate contains 400,000 samples in 10 s. Because the effective depth of a recurrent network when unrolled over time is equal to the number of time steps multiplied by the number of hidden layers, a long sequence will essentially result in a prohibitively deep network, which makes training infeasible (note that for our problem the signal sampling rate is quite small, so training is still feasible). Consequently, deep learning researchers have developed alternative convolutional architectures that work well with sequence data. A highlighted example is the WaveNet architecture that was developed and widely used by Google for speech processing and synthesis [45].

For that reason, we also tested two variants of convolutional networks: a shallow convolutional neural network (CNN) and a WaveNet architecture. The shallow CNN consists of one convolutional layer (256 filters, kernel size: 6, stride: 1) followed by two fully-connected layers (128 and 2 cells, respectively). We used a drop-out rate of 0.4 for all layers. The main components of the WaveNet architecture are residual blocks with residual connections between them. Within each block, there are several layers of dilated convolutions coupled with gated mechanism to learn what important features in the time series it should focus on. The dilated convolution is similar to a regular convolution except that the convolution kernel will skip the input by a certain number of steps. This helps to double the receptive field size of each layer, which enables the network to learn long-range dependency in the sequence. In this study, we used two residual blocks and three dilated convolution layers within each block (32 filters, kernel size: 6, stride: 1). As before, we used a drop-out rate of 0.4 and a final 2-cell fully-connected layer as the output. Further, we do not use the skip connection from the input to the residual blocks, as proposed in the original WaveNet architecture [45]. Note that all of the above hyperparameter values were chosen based on extensive cross-validation on the public dataset.

#### 2.3.3. Performance Metrics

For model training, we used step classification accuracy as the validation metric. It indicates the percent of steps classified correctly as left step or right step and is computed as follows:(1)$$accuracy_{step\_classification}=\left(\frac{n_{correct}}{N_{total}}\right)\times 100$$
where $n_{correct}$ is the number of correctly classified steps and $N_{total}$ is the total number of steps.

The second metric is step-count accuracy. Based on the sequence of left/right steps, each step is defined as the transition from left to right or vice versa. Therefore, we can compute the ground truth step count from the annotated labels and similarly for the model prediction. Given that, the step-count accuracy was computed as follows:(2)$$accuracy_{step\_count}=\left(1-\left|\frac{step\_count_{predicted}-step\_count_{ground\_truth}}{step\_count_{ground\_truth}}\right|\right)\times 100$$
where $step\_count_{predicted}$ is the number of steps computed from the model’s prediction, and $step\_count_{ground\_truth}$ is the number of steps computed from the ground truth annotation.

#### 2.3.4. Model Training and Validation on Public Dataset

To prepare the data for model training, we used a sliding window over both the acceleration signal and the step type. The sliding window has a step size of 1. A noteworthy difference compared to the previous study [38] was the way we computed the label for training. The general problem is that, given a window of acceleration, we need to define the step type of that window (i.e., left or right step). However, because the window may contain both left and right steps, it can be tricky to determine the step type of a window. That issue can be aggravated by a long window because it may contain multiple steps. The previous study chose the most frequent step type as the label of a window [38]. Because of the issue mentioned above, the chosen window size was pretty short in that study (0.46 s). To alleviate that problem, we chose the step type of the last time step as the label of a window. In other words, we rephrased the problem as predicting the current step type (left/right) based on past sensor signals. That enabled us to use an arbitrarily long window size so the neural networks could learn potential long-range dependencies in the sensor signals.

Cross-validation on the public dataset was done by splitting the subjects into training (28 subjects) and validation (2 subjects) sets. We performed 70 shuffled iterations of cross-validation for each model. We implemented the neural networks using Keras with TensorFlow as the backend. Based on extensive cross-validation, we selected the best hyperparameters as follows: the window size was 2 s (30 time steps) for the recurrent network and 4 s (60 time steps) for the convolutional networks. Note that the optimal window size was 4–8 times longer than in the previous study. This suggests our choice of label (the last step type instead of the most frequent step type) allowed the neural networks to handle a longer window. The batch size was 256 for the convolutional networks and 512 for the recurrent network, the optimizer was Adam with default values (learning_rate = 0.001, beta_1 = 0.9, beta_2 = 0.999, epsilon = $1\times 10^{-7}$, amsgrad = False), the loss function was categorical cross entropy, the number of epochs was 50 and the early stopping had a patience of 10. Note that for early stopping, we chose validation accuracy as the metric to monitor instead of validation loss because that was more relevant to our optimization goal.

#### 2.3.5. Model Testing on Independent Dataset

To test the models on the independent dataset, we chose the best models for each architecture. Note that although the models’ prediction was left or right step given a sample window, we focus exclusively on step-count accuracy because that is the measure of interest here.

For the first approach, we ran the best pre-trained models on the accelerometer data in the AZ dataset to derive the step count for each subject (Figure 4b). Essentially, this approach is the ultimate test for the generality of the models trained on the public dataset because there was no further tuning of the parameters. Therefore, the generalized approach would test whether the neural networks could work well on the data from completely different devices, subjects and testing environments.

For the second approach, we performed domain adaptation on individual subjects before deriving step counts (Figure 5). Specifically, we trained the general models on 30 s of labeled data for each subject and used that to derive the step count. In other words, we had a personalized model for each subject. The advantage of this approach is that it can adapt to the particularities of each subject’s walking gait. However, it requires training labels for each subject, which can become prohibitively expensive for a large number of subjects. To mitigate this problem, we limit the domain adaptation to a very short training window.

## 3. Results

### 3.1. Cross Validation on Public Dataset

We trained and cross-validated three neural network models (recurrent network with LSTM cells, convolutional neural network and WaveNet) on a public dataset consisting of acceleration signals from 30 subjects performing regular walking. An extensive search over the hyperparameter space was conducted to find the best configuration for each architecture based on the cross-validation results. All the cross-validation results presented below were taken from 70 shuffled splits of training and validation sets.

Figure 6 shows the cross-validation accuracy of the best models for both step classification and step count when we used the raw acceleration as input to the neural networks. As a reference, we indicated the performance achieved in a previous study [38] with a dashed blue line (∼60% for both tasks). In the step classification task, all neural network architectures performed relatively well and similar to each other (Figure 6, left panel). The mean accuracy is 78.84% for CNN, 79.19% for WaveNet and 80.14% for LSTM. We also found relatively small variability across the splits of training and validation sets, which suggests the models are quite robust with regard to different splits of training data. The right panel of Figure 6 shows step-count accuracy for the neural networks. Overall, step-count accuracy is very high and similar across all models. The mean step-count accuracy is 98.32% for CNN, 98.36% for WaveNet and 98.4% for LSTM. Moreover, the variability across different training–validation splits is also very small.

As demonstrated in Figure 2, the raw acceleration signals might vary substantially across different devices and wear locations. Therefore, we also performed training and validation on the ENMO measures derived from raw acceleration. By using the vector magnitude instead of the raw acceleration, we expect the variation across devices and/or wear location to be less troublesome (see Figure 3 for an illustration). Similar to the raw acceleration results, all neural network architectures performed relatively well and similar to each other in the step classification task (Figure 7, left panel). The mean accuracy of step classification is 77.59% for CNN, 76.64% for WaveNet and 79.43% for LSTM. We also observed larger variability compared to the raw acceleration results. Although all models performed relatively well on the step-count task, convolutional networks tended to be better (CNN: 98.72%, WaveNet: 98.51%) compared to LSTM (94.87%) (Figure 7, right panel). Therefore, the LSTM model seems to be worse with the ENMO signals compared to when using raw acceleration data.

Overall, the results suggest that the neural network models are quite robust with regard to the type of input (i.e., raw acceleration vs. ENMO). At a closer look, it seems that the models work slightly better with raw acceleration data. The reason may be the raw acceleration data contains more information than the ENMO measure. Another observation is that the convolutional architecture (CNN and WaveNet) tends to be better and more robust than the LSTM. This is especially true for the simplest model, CNN.

The fact that step-count accuracy is significantly better than step classification accuracy was also found in a previous work [38]. At first glance, the result may be puzzling because we did not train or validate the model directly on step count but only used the step classification (i.e., left or right) for the training. However, it is expected because step classification is noisier and less robust than step count. For example, if there is misalignment of the sensor time series and the step type label, step classification will be significantly degraded while step count should be almost the same. Therefore, the results suggest that by training the neural networks on step classification, they could learn a robust representation that is relevant for step count.

### 3.2. Test of General Models on Independent Dataset

To test the generalizability of neural network models, we chose the best pre-trained models based on the above cross-validation results and tested them directly on the data collected at AstraZeneca. First, we look at the results using the raw acceleration as input into the models. The step-count accuracy for each subject, device and model is shown in Figure 8. For comparison, we also ran the models on the five subjects reported in the previous work [38] (indicated as blue dots in Figure 8). In that study, only the Actigraph wearable was used, so we can only make a comparison for that device. Moreover, most devices (Actigraph, Apple Watch and iPhone) also have a built-in pedometer, thus we also included the performance of those native algorithms in the comparison. As a summary of model performance, the mean accuracy was tabulated in Table 1. For the Actigraph device, the first noticeable observation is that the built-in algorithm performs poorly compared to our models, with mean accuracy of 79.56%. All three neural network architectures work very well, with the mean performance reaching 98% for CNN and LSTM models while WaveNet had an outlier subject that degraded its performance to 91.41%. Furthermore, the results across all models are quite similar for subjects in our study and those in a previous study [38]. For the Shimmer device, the performance of all neural networks was worse than Actigraph, and the worst architecture is the LSTM network (8.04%). In contrast, the LSTM performs very well for the Apple Watch (95.65%), while both CNN and WaveNet see signficant decreases in performance. In addition, the built-in algorithm of the Apple Watch works very well, with a mean accuracy of 98.2%. We observe a similar pattern for the iPhone. The best neural network architecture is the LSTM network (96.93%), while the CNN and WaveNet have worse performance. We also observe high performance from the built-in algorithm of the iPhone (98.2%). It is sensible to see similar patterns from the Apple Watch and iPhone because they come from the same vendor and presumably both the sensors and the algorithms should be relatively similar. In general, we found that neural network performance varied substantially across devices and subjects. Furthermore, in some cases, the neural networks performed well for all subjects except for one outlier (e.g., WaveNet for the Actigraph and Shimmer, and CNN for the Apple Watch and iPhone). This could be due to substantial variations in the signals from device sensors (as discussed above and illustrated in Figure 2). Therefore, in the next section, we will look into the models’ performance using ENMO instead of raw acceleration because it is supposed to be less sensitive to the devices’ sensors (see Figure 3).

Figure 9 shows the models’ performance for individual subjects when we used ENMO as the input. The mean step-count accuracy for each device and model is also shown in Table 2. Overall, we see substantial improvement for all models across the devices. Results for the Actigraph show the least improvement because the models were already very good with raw acceleration. However, we see a significant boost for the WaveNet model (from 91.41% to 98.82%), mostly because it can better handle the outlier subject found with raw acceleration. The Shimmer device sees the largest improvement. We found high accuracy across all models (∼96%). The performance boost is most obvious for LSTM because its accuracy with raw acceleration is only 8.04%, as we saw above. Although we see a clear outlier subject that is worse than the others, the accuracy for that subject is still around 80%. For the Apple Watch, we also observed a significant increase in the performance of CNN and WaveNet models, with both approaching 96% accuracy (compared to 74–80% with raw acceleration). In contrast, the LSTM model has a slightly worse performance with ENMO compared to raw acceleration (94.14% vs. 95.65%). We found a similar pattern for the iPhone. Both the CNN and WaveNet gain a substantial improvement in performance by using ENMO instead of raw acceleration (8–14% increase in accuracy). However, we see a drastic reduction in accuracy for the LSTM model, with a drop from 96.93% down to 60.44%. Therefore, the results seem to be consistent between the Apple Watch and iPhone.

In general, we see a substantial improvement in the models’ performance by using ENMO instead of raw acceleration. It may be because ENMO is less sensitive to the specificity of device sensors compared to the raw acceleration. One exception from our results was the LSTM model for the Apple Watch and iPhone. The LSTM model consistently performs better on raw acceleration compared to ENMO for these Apple devices.

Among the tested devices, the iPhone is the most challenging case because it is the most different from the device used in the training dataset, both in device type (mobile phone vs. wearable) and wear location (pant pocket vs. wrist). As a result, the models (except CNN) still struggle to perform well for the iPhone, even when we used ENMO.

### 3.3. Test of Personalized Models on Independent Dataset

Although the convolutional neural networks perform well with ENMO, we still see outlier subjects in some case, most notably for the Shimmer device. Those cases could be important in clinical applications because the patient population is likely to exhibit idiosyncrasies in their walking gait and render the general models inaccurate in step-counting task. Therefore, we also tested another approach—domain adaptation—where we trained the pre-trained models on a small subset of labeled data for individual subjects and used those personalized models to derive step count.

Similar to the previous section on general models, we started with domain adaptation on raw acceleration input. Figure 10 shows model performance for individual subjects across all devices, and Table 3 shows the mean accuracy for each model and device combination. Compared to the general models (Figure 8 and Table 1), the personalized models show significant improvement in most cases. Specifically, the CNN model maintains similar performance for the Actigraph and gets a substantial boost in performance for the other devices (10–24% increase in accuracy). Likewise, the WaveNet model sees considerable improvement across all devices (7–27%). For the LSTM model, we also see an improvement in performance for the Shimmer, Apple Watch and iPhone, with the most drastic change corresponding to the Shimmer (41% increase in accuracy). However, the LSTM model has worse performance for the Actigraph (17% decrease in accuracy). This reduction in the performance of the LSTM model is due to two outlier subjects with very low accuracy. We also observed the same pattern for the Shimmer device.

Next, we tested the domain adaptation approach for ENMO input. Model performance for individual subjects is shown in Figure 11, and the mean step-count accuracy is shown in Table 4. Similar to the results for domain adaptation on raw acceleration, we found an overall improvement for all models and devices compared to the performance of general models (Figure 9 and Table 2). The CNN model sees an improvement across all devices although the accuracy increase is generally small because the general CNN model already works well with ENMO input. We see a similar pattern for the WaveNet model, with a consistent improvement across devices, especially the iPhone (10% increase in accuracy). The LSTM model observes a minor improvement for the Actigraph and Shimmer, a slight decrease in performance for the Apple Watch and a marked improvement for the iPhone (33% increase in accuracy). Notably, the domain adaptation method using ENMO does not contain drastic outliers as seen in other approaches. The results demonstrate that this approach can handle the outlier subjects found in our study.

## 4. Discussion

In this study, we developed neural network algorithms for step counting and conducted extensive testing of two transfer learning methods (generalization and domain adaptation) on four devices. They exhibited state-of-the-art performance on an independent test set. Importantly, these algorithms were designed with special considerations for the critical requirements of clinical trials. Firstly, they only used accelerometer data (and not gyroscope data), which significantly reduced battery consumption, prolonged the use time without charge and, hence, may improve patients’ wearing compliance. In addition, that may result in higher integrity of recorded data, which is of paramount importance in clinical trials. Second, it is device agnostic and highly accurate for both wrist-worn and pocket devices. Since patients in clinical trials can be very different from healthy subjects, black-box built-in algorithms from the manufacturers are not appropriate because we may have to adapt and fine-tune the algorithms for the disease-specific population.

There are a few notable observations from the study’s findings. First, although raw acceleration input resulted in better model performance during cross-validation (see Figure 6 and Figure 7, ENMO often gave better results when tested on a different device (see Figure 8, Figure 9, Figure 10 and Figure 11). The results are generally concordant with research in neural networks, which shows that, although they can learn well from raw data, feature engineering often leads to significant performance improvement. It suggests that when we aim to obtain better generalization across devices, ENMO measures should be used because they are less sensitive to the specifications of device sensors. Moreover, future research into the utility of combining raw and other engineered features such as ENMO is warranted. Second, it is quite consistent across different approaches that convolutional architectures (CNN and WaveNet) performed better than the LSTM network, especially in domain adaptation using raw acceleration data. Both CNN and WaveNet worked very well for all subjects, whereas LSTM worked poorly for a few outliers. In addition, the training process for LSTM was also often less stable than convolutional networks. The findings are generally consistent with recent research showing superior performance of convolutional vs. recurrent architecture [44]. Third, the simple CNN model has consistently demonstrated its superior performance as well as robustness compared to other models, especially when using ENMO as input. The reason could be that complex models such as WaveNet and LSTM might have overfit during training (even with cross-validation) while the simple CNN model strikes the right balance between flexibility and robustness. The results may have significant implication in situations where algorithms deployed on the mobile device itself are limited by computational resources.

We see significant variations in the built-in algorithm of the devices, ranging from the worst (Actigraph) to the best (Apple Watch, iPhone). This illustrates the need for standardization and validation of pedometer devices, especially those used in clinical settings. Surprisingly, the most-popular device used in clinical application, Actigraph, performed poorly, while the popular consumer devices (Apple Watch and iPhone) were highly accurate. These Apple devices were better than the general models and were only slightly worse than the personalized models. Note that these built-in pedometers were optimized for each device, so they are not directly comparable to the general models. Furthermore, we may still need an approach such as domain adaptation to handle the outlier cases likely in a patient population.

Finally, we want to comment on the operational aspect of deploying the algorithms in practice. The results demonstrated a popular trade-off between general models and personalized models. On the one hand, the general models are simpler to deploy in a clinical trial because the same model is applied to all subjects. However, it may not be able to handle outlier subjects, which could be important in clinical contexts. On the other hand, the personalized models resulted in superior accuracy even for outlier subjects. Nevertheless, they are not easily applied at scale because of the expensive manual labeling. Therefore, we suggest that in practice a hybrid approach could be employed. Specifically, we can have a short calibration process at the beginning for each subject in order to test the accuracy of the general models. The calibration would consist of a short walk during which the subject counts his/her steps to provide feedback for the general models. Importantly, subjects could do this on their own. Afterwards, we would only use the personalized method for those subjects that the general models could not handle. With that, we can still obtain high accuracy for all subjects while limiting the expensive manual labeling of the personalization process (assuming there are not many outlier subjects).

## Figures and Tables

**Figure 1 sensors-22-03989-f001:**
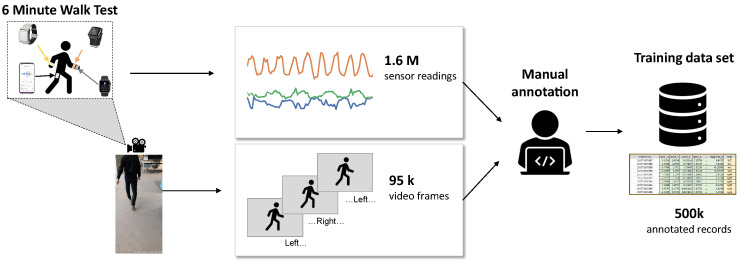
Overall workflow to generate the AZ dataset.

**Figure 2 sensors-22-03989-f002:**
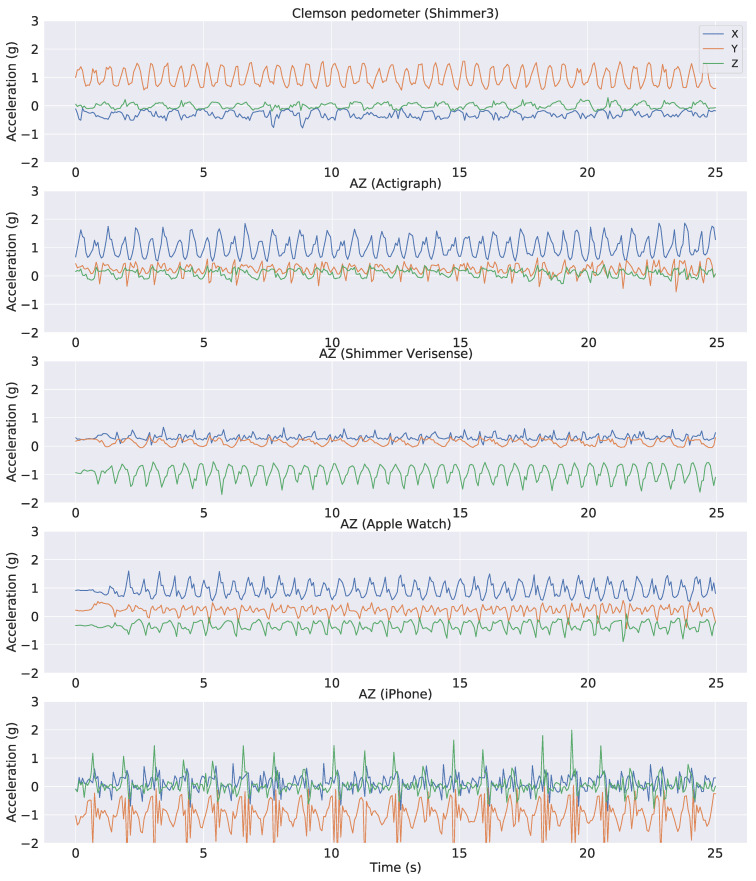
Representative raw accelerometer data from public and independent datasets. The first panel indicates data from the public Clemson dataset. The other panels illustrate AstraZeneca dataset for different devices used in the study.

**Figure 3 sensors-22-03989-f003:**
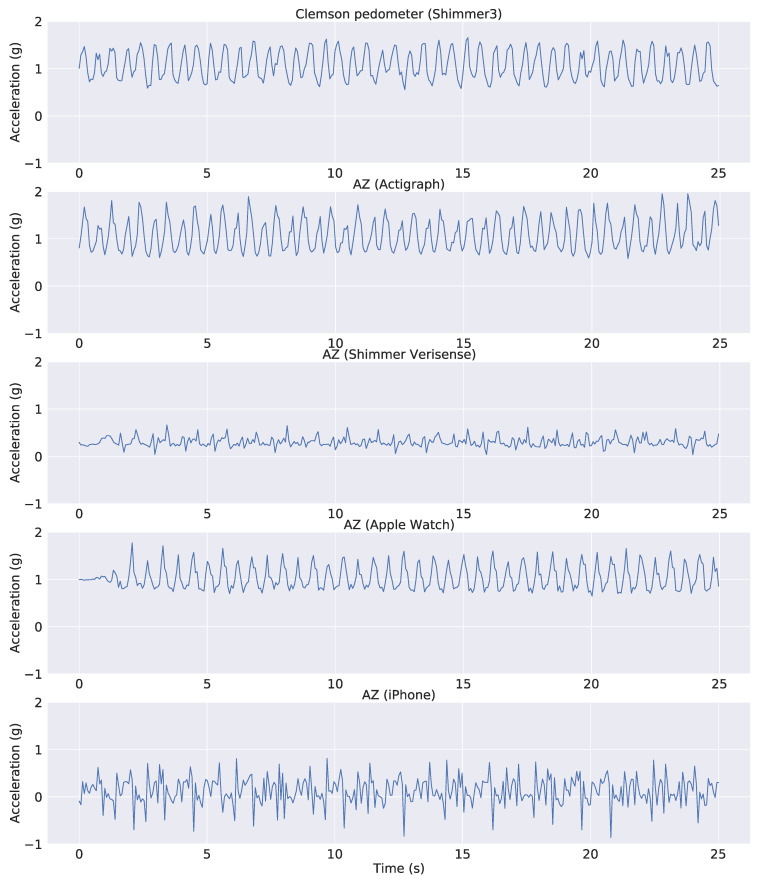
Representative ENMO signals from public and independent datasets. The first panel indicates data from the public Clemson dataset. The other panels illustrate AstraZeneca dataset for different devices used in the study.

**Figure 4 sensors-22-03989-f004:**
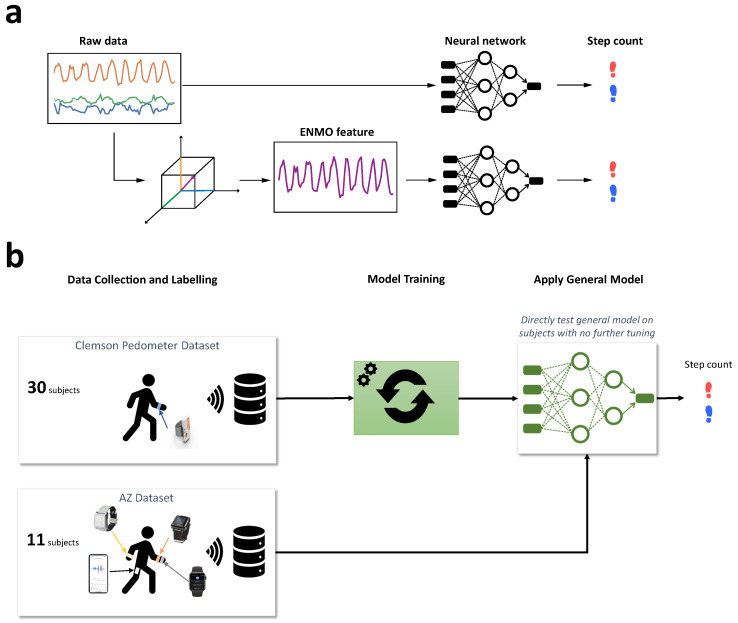
Workflow of training and testing step-count algorithms. (**a**) Model input was either the raw acceleration signal or the ENMO feature; (**b**) Generalized transfer learning on test data.

**Figure 5 sensors-22-03989-f005:**
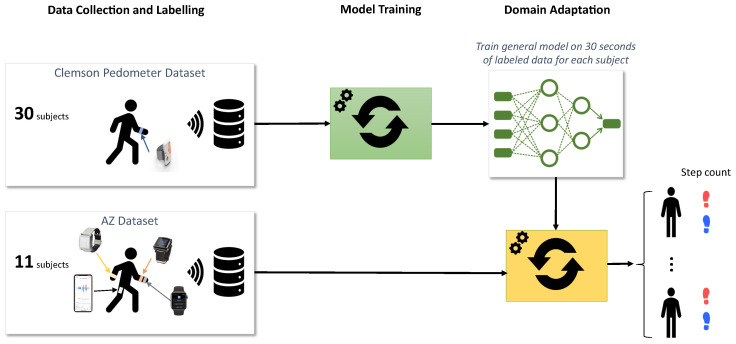
Personalized transfer learning on test data.

**Figure 6 sensors-22-03989-f006:**
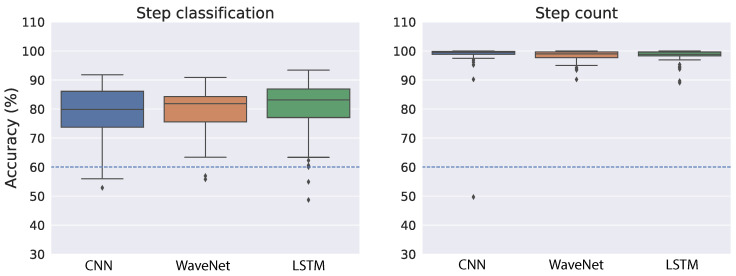
Cross-validation results on the public dataset using raw acceleration as the input to neural network models. The accuracy measures were taken from 70 shuffled splits of training and validation sets. The dashed blue line indicates performance achieved from a previous study [38].

**Figure 7 sensors-22-03989-f007:**
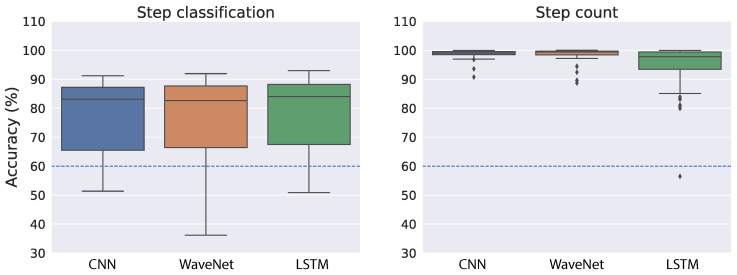
Cross-validation results on the public dataset using ENMO as the input to neural network models. The accuracy measures were taken from 70 shuffled splits of training and validation sets. The dashed blue line indicates performance achieved from a previous study [38].

**Figure 8 sensors-22-03989-f008:**
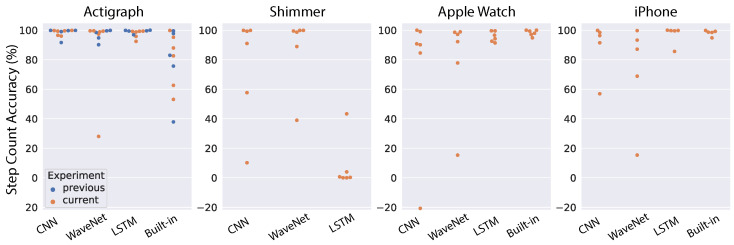
Results of testing the general models on an independent dataset using raw acceleration. Each dot correspond to one subject (blue dots indicate data in [38]).

**Figure 9 sensors-22-03989-f009:**
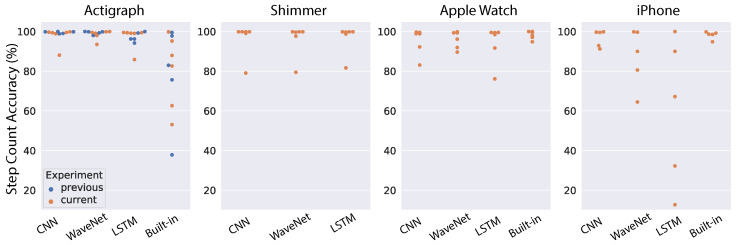
Results of testing the general models on an independent dataset using ENMO. Each dot correspond to one subject (blue dots indicate data in [38]).

**Figure 10 sensors-22-03989-f010:**
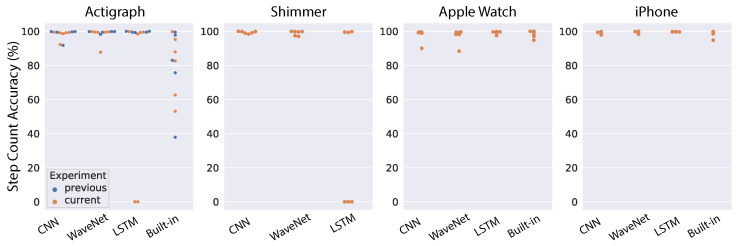
Results of testing the personalized models on an independent dataset using raw acceleration. Each dot corresponds to one subject (blue dots indicate data in [38]).

**Figure 11 sensors-22-03989-f011:**
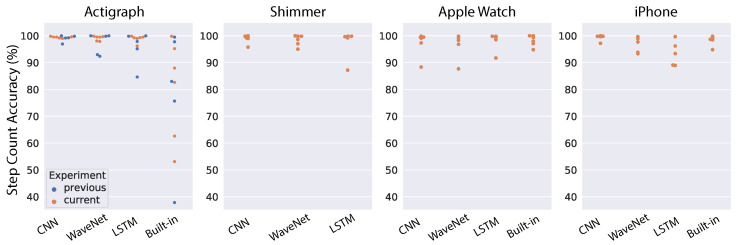
Results of testing the personalized models on an independent dataset using ENMO. Each dot correspond to one subject (blue dots indicate data in [38]).

**Table 1 sensors-22-03989-t001:** Step-count accuracy of testing general models on independent dataset using raw acceleration.

Test	Device			
**Algorithm**	**Actigraph**	**Shimmer**	**Apple Watch**	**iPhone**
CNN	98.28%	76.31%	73.96%	88.62%
WaveNet	91.41%	87.61%	80.01%	72.87%
LSTM	98.23%	8.04%	95.65%	96.93%
Built-In	79.56%	N/A	98.2%	98.2%

**Table 2 sensors-22-03989-t002:** Step count accuracy of testing general models on independent dataset using ENMO.

Test	Device			
**Algorithm**	**Actigraph**	**Shimmer**	**Apple Watch**	**iPhone**
Shallow CNN	98.46%	96.26%	95.4%	96.64%
WaveNet	98.82%	96.07%	96.04%	86.92%
LSTM	97.13%	96.62%	94.14%	60.44%
Built-In	79.56%	N/A	98.2%	98.2%

**Table 3 sensors-22-03989-t003:** Step-count accuracy of personalized models on independent dataset using raw acceleration.

Test	Device			
**Algorithm**	**Actigraph**	**Shimmer**	**Apple Watch**	**iPhone**
CNN	98.04%	99.35%	97.73%	99.19%
WaveNet	98.43%	98.97%	97.17%	99.53%
LSTM	81.39%	49.79%	99.25%	99.72%
Built-In	79.56%	N/A	98.2%	98.2%

**Table 4 sensors-22-03989-t004:** Step-count accuracy of personalized models on independent dataset using ENMO.

Test	Device			
**Algorithm**	**Actigraph**	**Shimmer**	**Apple Watch**	**iPhone**
CNN	99.26%	98.92%	97.26%	99.3%
WaveNet	98.15%	98.39%	96.9%	96.7%
LSTM	97.33%	97.55%	98.11%	93.45%
Built-In	79.56%	N/A	98.2%	98.2%

## Data Availability

The Clemson Pedometer dataset can be found here: http://cecas.clemson.edu/~ahoover/pedometer/ (accessed on 19 May 2022).

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
