# Peer review of "Accurate Step Count with Generalized and Personalized Deep Learning on Accelerometer Dataâ€"

_sensors, 2022, doi:10.3390/s22113989_

Round 1
Reviewer 1 Report
It is a good quality manuscript. The step counting devices are a helpful tool to encourage people for physical activity especially during the covid-19 pandemic and temporary lockdown and/or isolation.
Author Response
Thank you for your positive feedback! We truly hope our works would contribute to that goal of encouraging physical activity among healthy and diseased population.
Reviewer 2 Report
Broad comments. The authors have made a concise overview of the topic and a concise reference to existing literature. They have indicated the main task of the paper among its motivation. Finally, they have pointed out the key message and the potential benefits of their work.
Specific comments. In general, the text is very well structured and has clearly defined topics. Some comments for improvement:
- As a general drawback, I could say that there is no reference to similar works that used wearables and deep learning methods (e.g. [1]) in different health monitoring areas:
[1] Daskalos A-C, Theodoropoulos P, Spandonidis C, Vordos N. Wearable Device for Observation of Physical Activity with the Purpose of Patient Monitoring Due to COVID-19. Signals. 2022; 3(1):11-28. https://doi.org/10.3390/signals3010002
- More or less all fundamental theory details that are needed are discussed and a review of the problem under evaluation is sufficient. It would be beneficial to clarify the reasoning and the behind the selection of the methods.
- Do authors think that the data used are statistically important?
- While in the discussion section authors describe the reasoning for deviations presented in table 4.
Author Response
Thank you for your constructive feedback. Please find below our point-by-point response to your comments.
As a general drawback, I could say that there is no reference to similar works that used wearables and deep learning methods (e.g. [1]) in different health monitoring areas:
[1] Daskalos A-C, Theodoropoulos P, Spandonidis C, Vordos N. Wearable Device for Observation of Physical Activity with the Purpose of Patient Monitoring Due to COVID-19. Signals. 2022; 3(1):11-28. https://doi.org/10.3390/signals3010002
Thank you for pointing to relevant literature. We included this in our reference for completeness.
More or less all fundamental theory details that are needed are discussed and a review of the problem under evaluation is sufficient. It would be beneficial to clarify the reasoning and the behind the selection of the methods.
We addressed the motivation for choosing neural network models instead of traditional methods in the introduction. In brief, neural networks have been successful in many tasks and do not require manual tuning like the traditional methods. We also employed a common technique, transfer learning, to assist with our model training.
Do authors think that the data used are statistically important?
Given the current sample size (5-11 subjects depending on the device), it is difficulty to perform statistical tests to determine the significance levels. Although the results are quite consistent across subjects, further works are needed to test our methods on a larger number of subjects and population.
While in the discussion section authors describe the reasoning for deviations presented in table 4.
In the discussion section, we did not refer explicitly to Table 4 but we pointed out that using ENMO instead of raw acceleration is an important component for the high performance presented in Table 4.
Reviewer 3 Report
In this work, the authors trained neural network models on publicly available data and tested on an independent cohort using 2 approaches: generalization and personalization. The general models may be simple to deploy in a clinical trial. Some suggestion are as follows.
(1) Please provide the detailed type, specification, and manufacturer of involved materials and devices.
(2) Please mark the characteristic peak in the data in Figures 2 and 3, so that the potential readers could know them fast.
(3) How are feature variables extracted? How to identify different actions using feature variables? The authors need to draw a diagram to elaborate on in the main text.
(4) The introduction of background is too long. The authors should simplify the related content, and highlight the key point and innovation.
(5) The text in some figures is hazy, such as Figures 8, 9, 10 and 11. The authors should improve their readability.
Author Response
Thank you for your constructive feedback. Please find below our point-by-point response to your comments.
(1) Please provide the detailed type, specification, and manufacturer of involved materials and devices.
We provided the details of all devices in section “2.1.2 Independent dataset”. For further clarity, we also included the versions of Apple Watch and iPhone tested.
(2) Please mark the characteristic peak in the data in Figures 2 and 3, so that the potential readers could know them fast.
Because our algorithms do not employ characteristic peak feature in the data, we think it may not add values to include that in the Figures 2 and 3. These figures were used mainly to illustrate the variation in acceleration signals across different devices.
(3) How are feature variables extracted? How to identify different actions using feature variables? The authors need to draw a diagram to elaborate on in the main text.
One advantage of using neural network models is that we don’t have to perform extensive feature extraction. Therefore, in this paper, we limited the model input to either raw data or the simple ENMO feature. The ENMO was computed using a simple formula provided in section “2.2 Data preprocessing”. Essentially, it is a Euclidian norm of the raw acceleration minus one.
(4) The introduction of background is too long. The authors should simplify the related content, and highlight the key point and innovation.
We have revised the introduction to simplify the context and highlight the key points of the paper.
(5) The text in some figures is hazy, such as Figures 8, 9, 10 and 11. The authors should improve their readability.
We retouched the figures to make sure all the text is clearer.
Round 2
Reviewer 3 Report
One concern remained. The authors need to draw a diagram to elaborate the working principle in the main text so that the potential could know it well.
Author Response
According to the reviewer's request, we added a diagram in Fig. 4a to illustrate the features used as inputs to the neural networks.
Round 3
Reviewer 3 Report
There is no other suggestion.